# Contribution of Multiplex Immunoassays to Rheumatoid Arthritis Management: From Biomarker Discovery to Personalized Medicine

**DOI:** 10.3390/jpm10040202

**Published:** 2020-10-30

**Authors:** Carlos M. Laborde, Patricia Castro-Santos, Roberto Díaz-Peña

**Affiliations:** 1Dreamgenics, Inc., 33010 Oviedo, Spain; carlos.martinez@dreamgenics.com; 2Faculty of Health Sciences, Universidad Autónoma de Chile, Talca 3460000, Chile; patricassan@gmail.com

**Keywords:** proteomics, rheumatoid arthritis, personalized medicine, biomarker, multiplex immunoassays

## Abstract

Rheumatoid arthritis (RA) is a multifactorial, inflammatory and progressive autoimmune disease that affects approximately 1% of the population worldwide. RA primarily involves the joints and causes local inflammation and cartilage destruction. Immediate and effective therapies are crucial to control inflammation and prevent deterioration, functional disability and unfavourable progression in RA patients. Thus, early diagnosis is critical to prevent joint damage and physical disability, increasing the chance of achieving remission. A large number of biomarkers have been investigated in RA, although only a few have made it through the discovery and validation phases and reached the clinic. The single biomarker approach mostly used in clinical laboratories is not sufficiently accurate due to its low sensitivity and specificity. Multiplex immunoassays could provide a more complete picture of the disease and the pathways involved. In this review, we discuss the latest proposed protein biomarkers and the advantages of using protein panels for the clinical management of RA. Simultaneous analysis of multiple proteins could yield biomarker signatures of RA subtypes to enable patients to benefit from personalized medicine.

## 1. Introduction

Rheumatoid arthritis (RA) is a multifactorial, progressive, systemic inflammatory autoimmune disease that affects approximately 1% of the population worldwide and primarily involves the joints, resulting in local inflammation and cartilage destruction [1]. Due to long-term disease complications, RA is associated with reduced life expectancy and constitutes a major cause of chronic incapacity.

Early RA diagnosis is essential for the improvement of prognosis and quality of life. In 2010, the American College of Rheumatology (ACR) and the European League Against Rheumatism (EULAR) established new RA classification criteria [2], which have been widely accepted by rheumatologists, that focus on features at an earlier disease stage. In addition to clinical criteria, they include two immunological markers: anti-citrullinated protein autoantibodies (ACPA) and rheumatoid factor (RF). RA can be classified as one of two forms: “early RA”, which affects patients in the first few years after clinical diagnosis, and “established RA”, which affects all RA patients with a disease duration greater than 1–2 years irrespective of the presence of clinical or radiological joint damage; however, there is no consensus as to the denomination of forms of RA [3]. The reality is that RA symptoms develop gradually. In the “preclinical phase” of development, there are relevant changes in the biomarkers ACPA and RF and another biomarker, anti-carbamylated protein antibodies (anti-CarP) [4]. The great interest in ACPA and RF stems from the fact that their expression may precede the clinical appearance of RA by 10 years or more, allowing for improved predictive accuracy when both ACPAs and RF are used simultaneously [5]. However, while progress has been made regarding the diagnosis of early RA, there is still a need for novel biomarkers to improve RA diagnosis. Treatment initiated at an early stage of the disease is more likely to be successful. Once RA is established, generally there will be damage to the joints and the occurrence of diverse comorbidities with complications as a consequence of chronic inflammation [6]. Early diagnosis and treatment prevent radiologic progression of RA and result in a higher rate of remission than delayed treatment [7,8]. The combination of multiple biomarkers improves the ability to represent complex pathophysiological conditions, which is a prerequisite for personalized treatment. In this review, we discuss the latest proposed biomarkers and the advantages of using panels of protein biomarkers in the clinical management of RA.

## 2. Rheumatoid Arthritis: A Complex Multifactorial Disease

A diagnostic phenotype of RA does not exist. RA is a disease in which the typical signs, such as pain and fatigue, occur before the onset of joint inflammation [9] and comorbidities may develop over time. The consequence is a heterogeneous disease with a variable clinical spectrum, although RA has been considered a relatively homogeneous clinical syndrome for more than 50 years. Like most rheumatic diseases, RA is a multifactorial entity in which genetic variability, environmental factors and random events interact to trigger activation of pathological pathways [10].

### 2.1. Genetic Factors

Rheumatoid arthritis-related autoimmunity initially develops years prior to the first joint symptoms in the genetically predisposed population as a result of a complex process that has not been elucidated for the most part. Moreover, these processes trigger different mechanisms that may be responsible for the different symptoms. This finding suggests that the mechanisms may differ between different genetic and/or immunologic subgroups of RA. Genetic susceptibility to RA is evident in familial clustering and monozygotic twin studies, and the heritability of RA has been estimated to be approximately 60% [11]. An improved understanding of the genetic basis of RA is required to develop a more personalized approach for disease treatment. The human “major histocompatibility complex, MHC” (HLA) region, located on chromosome 6, is by far the largest source of genetic susceptibility to RA. The region consists of genes encoding molecules that play a key role in the immune system by modulating responses to invading pathogens. Approximately one-third of genetic risk is associated with the HLA locus [11], specifically with the HLA-DRB1 shared epitope (SE), which encodes a common amino acid sequence associated with RA susceptibility and progression (^70^QRRAA^74^, ^70^QKRAA^74^, or ^70^RRRAA^74^) [12]. HLA-DR loci encode HLA class II molecules (DR, DQ and DP) that function as receptors for processed peptides derived predominantly from membrane and extracellular proteins. HLA-DR expression occurs on the surface of antigen-presenting cells (APCs) (including dendritic cells, macrophages and B cells) and is essential for displaying peptides to CD4+ T cells, inducing their activation. The presence of an SE suggests that the HLA alleles bind the same antigen, suggesting the existence of an arthritogenic self-peptide or molecular mimic of foreign antigens that shapes the T-cell-antigen repertoire. However, this arthritogenic peptide has not yet been identified.

The pathogenesis of RA has a polygenic basis, and genetic variation can be explained by RA risk alleles within the non-HLA locus. In the last few years, our understanding of the genetic susceptibility and pathogenic mechanisms underlying RA has been greatly improved by the increase in the power of genome-wide association studies (GWAS). These studies have uncovered over 100 genetic loci associated with an increased risk of RA, establishing 98 genes that could potentially contribute to the onset of RA [13]. HLA and some non-HLA genes have been linked to the development of antibodies against citrullinated proteins, allowing differentiation between ACPA-seropositive and ACPA-seronegative RA [13,14]. However, analyses of ACPA-negative RA over the years have been challenging. Indeed, it has been suggested that seronegative RA may be heterogeneous with regard to genetics and related pathogenic mechanisms [15].

### 2.2. Environmental Factors

There are other factors that provide additional complexity and need to be considered. In Table 1, we show the most important of these factors. Smoking is the most common environmental risk factor for RA, increasing the risk of disease by 1.3- and 2.0-fold for women and men, respectively [16]. This finding has led to a model of pathogenesis in which smoking exposure may lead to the presence of citrullinated proteins in the lungs, which is followed by the development of ACPAs in susceptible individuals [17]. Considering that it likely reflects the action of environmental exposure, the presence of ACPA in an unaffected individual is clearly a risk factor for the future development of RA. Exposure to air pollution has been investigated, providing variable results [18]. There seems to be an increased risk of RA development in individuals living close to roads with heavy traffic [19]. Additionally, occupational exposure to mineral oils and silica has been associated with an increased risk of developing RA [20,21].

Although the disease can develop at any age, the typical clinical presentation of RA first occurs between 20–40 years of age, affecting women more frequently than men at a 2–3:1 ratio [1]. A hormonal influence on the onset of the disease in women has been suggested. In fact, pregnancy is in itself a known risk factor for the development of RA [22,23]. However, the mechanism by which gender influences susceptibility to RA remains unclear. Additionally, the role of diet and exercise requires more study before the proposal of relevant recommendations [18]. What seems clear is that most studies have demonstrated an increased risk of RA in individuals with obesity [24]. In addition, studies of bacterial communities in the intestine and the lung have revealed the microbiome as an emerging area of interest in RA [25,26], but no distinct features of the gut or lung microbiomes have been observed prior to the development of RA so far. With regard to bacterial infections and the risk of RA, more tangible results have been obtained for periodontitis-associated bacteria [27]. Other potential risk factors include several physical and psychosocial aspects [28], the most prominent of which is socio-economic status [29].

## 3. Serum Biomarkers in Rheumatoid Arthritis

Early diagnosis and immediate, effective therapy are crucial to gain control of inflammation and prevent deterioration, functional disability and unfavourable progression in RA patients. Much of the current pharmacotherapy is implemented on a “trial-and-error” basis [30]. All of the RA-associated symptoms, in addition to having serious medical consequences for the individual patient, also economically affect the healthcare system and society as a whole. To carry out personalized medicine for RA, clinical practice requires the use of biomarkers to ensure early diagnosis, accurate stratification and the high efficacy of treatment. A very large number of biomarkers have been investigated in RA, although only a few have made it past the discovery and validation phase into clinical use [31]. RF and ACPA are currently used as biomarkers for diagnostics and can predominantly be detected in the serum and synovial fluid (SF) of RA patients [32] (Table 2). RF has moderate specificity because it can also be found in other connective tissue diseases and in some chronic infectious diseases [33]. ACPA is more specific, and the presence of ACPA is an independent risk factor for developing RA in patients with undifferentiated arthritis [34]. However, in reality, RA constitutes two clinical RA subgroups, ACPA-positive and ACPA-negative RA, that differ with respect to genetic background, predisposing environmental factors and clinical progression/remission [35]. The pathogenic significance of ACPA in RA may be the result of its multiple biological activities [36]. The production of ACPA disrupts immune tolerance and is dependent on the occurrence of both genetic and environmental factors. In a hypothetical model, environmental factors could contribute to stimulating innate immunity. Apoptosis and/or necrosis of some cells could cause citrullination of certain proteins in the lungs (due to the increase in the activity of peptidyl-arginine deiminase (PAD) enzymes). Some of these modified proteins bind specifically to human leukocyte antigen (HLA)-DR molecules on antigen-presenting cells (APCs), resulting in high titres of ACPA. Citrullination of proteins in the joints due to infection, trauma, exercise, etc., could lead to immune complex formation between the modified proteins and ACPA, which could further bind to Fc receptors on the surface of synovial macrophages, contributing to the perpetuation of inflammation. The erythrocyte sedimentation rate (ESR) and C-reactive protein (CRP) are used as markers of inflammation and are significant indicators of disease severity [37]. However, these markers only reflect the level of inflammation, and their use is limited.

In Table 2, selected promising biomarkers are presented. Anti-CarP antibodies have been identified in both ACPA-positive and ACPA-negative RA patients in sera and are also linked to more severe disease [38]. Carbamylation is a post-translational modification induced by the presence of cyanate in which the enzyme lysine carbamyltransferase catalyses the carbamylation of a lysine residue into homocitrulline. Compared to ACPA and RF, anti-CarP antibodies show lower sensitivity for the diagnosis of RA [44]. However, a triple-positive autoantibody profile (RF, ACPAs and anti-CarP) provided promising results that might help to identify individuals at risk of developing RA [45]. Other antibodies of growing interest in RA target PAD enzymes. These enzymes catalyse the conversion of peptidyl-arginine into peptidyl-citrulline, a posttranslational modification known as citrullination. In particular, anti-PAD3 and PAD4 antibodies have emerged as significant participants in the pathogenesis of RA [46]. Ren J et al. reported a meta-analysis showing the high value of anti-PAD4 in the diagnosis of RA and the high specificity but relatively low sensitivity of anti-PAD4 [39]. Moreover, anti-PAD4 antibodies have been associated with severe erosive joint disease in RA [40]. Recently, anti-PAD2 has been described for the first time in RA [47]. Anti-PAD2 seems to be characteristic of a clinically distinct subtype of RA causing less severe baseline joint inflammation and slower joint disease progression [47]. Anti-mutated citrullinated vimentin (anti-MCV) antibodies have demonstrated comparable diagnostic value to RF and ACPA and might be an effective diagnostic marker for RA [41]. An association between anti-MCV levels and extra-articular manifestations of RA has been detected [48]. Moreover, anti-MCV seems to also be a predictive marker of the response to rituximab treatment [49]. Several antibody types have been identified only in certain subsets of RA and present a complex pattern with partial overlaps between the different antibodies. Antibodies to stress-related proteins [50] and modified type II collagens [51] and their role in RA have also been explored. Alterations in the glycosylation status of antibodies are associated with autoimmunity [52]. Interestingly, glycosylation of the Immunoglobulin G (IgG) ACPA V domain and its value as a predictive marker of RA development in ACPA-positive individuals have been reported [53].

Serum 14-3-3η protein has been considered a promising joint-derived biomarker for RA [42,54]. Combined with other serological measurements, detection of the 14-3-3η protein enhances the diagnosis of RA, especially in patients seronegative for RF and ACPA [42]. A positive 14-3-3η status has also been significantly associated with radiographic progression in early RA at years 1, 3 and 5, indicating the prognostic utility [55]. A recent meta-analysis demonstrated that serum and synovial calprotectin are correlated with disease activity in RA [43]. Furthermore, calprotectin plays a potential role as an independent predictor of radiological progression [56].

Proteomics research focuses on the discovery of novel biomarkers, which would not only provide an early diagnosis and allow us to understand the different pathological mechanisms underlying the heterogeneity of RA but also would allow to stratify patients, which is critical to enable effective treatments [57]. Patients show a huge variability in response to different treatments, and some of them are exposed to adverse effects. For example, although anti-tumour necrosis factor (TNF) α therapy is effective in the majority of cases, 30 to 40% of treated patients show an inadequate response [58]. Several approaches have shown the advantage of using proteomics to identify predictors of treatment response in RA [59], mainly with the TNF blockers etanercept and infliximab [60,61,62,63,64]. Biomarkers for prognosis of RA would help identifying the patients with the highest progression rates which will allow focusing on them with more aggressive treatments. It has been a long time, since Hueber et al. last described in 2005 an autoantibody signature with the potential to stratify patients by disease severity [65]. They analysed serum samples with antigen microarrays, showing potential to stratify RA patients with less disease severity. However, the search for predictors of prognosis and/or treatment response in RA patients remains an area of development. More efforts need to be made in order to discover new biomarkers, validate the findings and translate them into the clinical practice.

## 4. Multiplex Immunoassays: A Promising Path to Personalized Medicine

Proteomics dynamically study the set of proteins expressed by an organism at a given time and under certain specific conditions of time and environment [66,67]. This type of studies can be approached with different techniques, such as gel-based and mass spectrometry-based techniques, single-molecule sequencing and multiplexed immunoassays. All of them are complementary and valid techniques to study the proteome, with their particular advantages and disadvantages that condition their applicability. Despite all of them being equally interesting, in this review, we have focused on multiplexed immunoassays. We trust that they have the greatest potential for their application in clinical practice and will be the most interesting for readers.

### 4.1. Limitations of Current Proteomic Technologies

Proteomic research reveals hundreds of candidate proteins every day around the world. However, despite this large number of proteins, the number of candidate protein biomarkers that eventually reach clinical use remains very low. In some cases, proteins have either unknown or incompletely characterized clinical significance, while other proteins do not exhibit expression that is consistent or sufficient enough to warrant their use in clinical practice [68,69,70].

Any proposed biomarker should be of clinical interest. This implies that its role in the disease must be well established; it must be measurable in biological fluids; it must result in improved sensitivity and specificity compared to existing biomarkers, and its detection must be cost-effective. Plasma and urine are the most commonly used biological samples for laboratory tests. Only in special cases are extravascular fluids, such as cerebrospinal fluid or synovial fluid, used. Blood is the gold standard because it travels through almost every part of the human body, and therefore, serum/plasma biomarkers are particularly valuable for monitoring health status [71]. Apart from blood, urine is a very interesting sample type for studying secreted proteins, and it can be obtained by non-invasive techniques, which makes it especially attractive in some cases. Research projects that identify candidate proteins using other types of human biological samples (homogenized tissue, cell supernatants, etc.) must confirm two things: that the proteins are measurable in plasma or urine and that the same alterations previously observed between healthy controls and patients are corroborated in one of those two types of specimens.

The direct discovery of biomarkers in serum or plasma using proteomic techniques is not a simple task but a great challenge due to the complexity and high dynamic range of protein expression, which extends through eleven orders of magnitude, from albumin to cytokines [72]. Moreover, most candidate protein biomarkers are often present at very low concentrations and are often bound to other carrier proteins. In biological fluids, such as plasma and urine, high-abundance proteins are predominant and repress the signals of lower-abundance proteins, which then become undetectable by common proteomic techniques, such as two-dimensional gel electrophoresis and chromatography. Different commercial approaches have been developed predominantly to eliminate high-abundance serum proteins [73,74,75]. However, this required step involves altering the natural composition of the plasma and introduces a source of variability that can influence the validity of subsequent results. In addition, this step does not occur in clinical practice, so before considering the clinical application of candidate proteins, it is necessary to ensure that they can also be quantified directly in plasma using clinical analysers. The problem that may arise is that predominant, high-abundance serum proteins can significantly interfere with and influence the performance of the test as well as the quality of the result, resulting in the so-called serum matrix effect [76].

There is another problem derived from the limitations of proteomic techniques. Most of these techniques are essentially manual and time-consuming, unlike those used in clinical practice, where the entire analysis process is fast and automated. Many intermediate steps are necessary to study the low-abundance proteome (removing high-abundance proteins, using different buffers, etc.) and the higher the number of intermediate steps is, the larger the difference between the final sample and plasma is. In addition, since proteomic techniques are mainly manual, the final results depend greatly on the experience of the researcher, affecting the reproducibility of the results. This is unlike what occurs in clinical practice, where the analytical process is fully automated, and this guarantees very high reproducibility.

Overall, the limitations of proteomic techniques mean that the results must be carefully evaluated. The differences observed in the candidate proteins must be confirmed under clinical conditions. A complex analytical validation process to determine the sensitivity, specificity, linearity range, detection limit, etc., as well as the dependence of these parameters on the nature of the biological sample used and its integrity over time is also necessary [77]. Only if all of the drawbacks previously mentioned are successfully overcome will candidate protein biomarkers be able to be considered for clinical use.

### 4.2. The Challenge of Multiplex Immunoassays

Conventional multiplex immunoassays are conducted by incubating microbeads or a microarray with a biological sample, followed by the addition of a mixture of detection antibodies with the expectation that each detection antibody will bind to the target proteins bound to the corresponding capture antibody. While thermal agitation ensures that each reagent encounters its target, three major challenges affect the design and limit the number of proteins that can be used: antibody cross-reactivity, non-specific antibody binding and antibody interference [78] (Figure 1).

Cross-reactivity toward nontarget proteins is pervasive and widespread for antibodies [79,80]. It often occurs when proteins in the sample are structurally similar to the protein of interest. The antibody pair interacts with a protein other than the targeted protein. Cross-reactivity is arguably the biggest obstacle in establishing high-performance and large-scale multiplexed assays and requires that manufacturers carefully select and exhaustively test in-house-developed antibody pairs to ensure analyte specificity [81]. Nonspecific binding occurs when the antibody pair interacts with the sample container or other assay surfaces and contributes to the background. This interference can be reduced by blocking. Finally, antibody interference occurs when endogenous antibodies within samples cross-link with assay antibodies or substances within the sample and prevent proper binding of the target protein to both the capture and detection antibodies. In biological samples, such as serum and plasma, common antibody interfering substances include human anti-mouse antibodies (HAMA) and RF as well as any substance present at an exceptionally high concentration [82,83].

Interfering factors can be found in complex sample matrices and may also be introduced to biological samples via reagents. Interference can result in either a reduced or elevated signal relative to the actual concentration of the target protein, yielding untrustworthy data. Therefore, interfering factors may compromise assay performance and design. Many protein combinations of interest may not be possible because nonspecific binding produces a large background signal, reducing the sensitivity. Although the main advantage of multiplex immunoassays is the possibility of quantifying several proteins simultaneously in a single test, a proper balance must be achieved to provide acceptable sensitivity and an appropriate dynamic range for each of the multiplexed proteins [84]. Consequently, the challenge of scaling up assays and enhancing their sensitivity requires that cross-reactivity is adequately addressed and suppressed or at least mitigated.

The appropriate composition of diluents and assay buffers represents another key factor in multiplex immunoassays. The diluent refers to the solution that is used for serial dilution of the protein standard to establish a calibration curve as well as the dilution of samples at concentrations above the upper limit of quantification. Thus, the composition of the diluent is an important factor that can affect the reliability and accuracy of immunoassays [85]. Ideally, the diluent must be optimized to maximize specific binding and detection, minimize nonspecific binding and interference, and mimic the sample matrix for the dilution of calibrators and samples [86]. Assay buffers should interact effectively with all reagents and proteins included under normal assay conditions. However, differences in hydrophobicity, electrical charge and post-translational modifications lead to the requirement for specific reaction conditions for each protein. Small changes in buffer pH or ionic strength can irreversibly change the structure, which can significantly affect the signal intensity and sensitivity of the assay [85].

### 4.3. Multiplex Immunoassay Platforms

Multiplex immunoassays use either antibodies or proteins/peptides as binding molecules to capture circulating proteins or autoantibodies, respectively, during incubation with biological specimens. Unbound proteins are removed by washing, and captured proteins are usually detected by using various labelled reporter ligands. After quantification of the detection label, the signal intensities can be converted into mass units using calibration curves or evaluated qualitatively [87,88].

Two different formats may be used: planar-based assays and microbead-based suspension assays (Figure 2). In the planar assay, the capture antibodies are immobilized on a two-dimensional support, and the fluorescent or chemiluminescent signals are identified. Planar arrays can be produced in two formats, which are either slide-based or microtiter-based. The slide-based format supports various layouts where repeated or individual assays composed of specific sets of antibodies are printed robotically upon the activated slide surface. Microtiter-based immunoassays involve antibodies within the wells of a standard protein-binding plate, similar to conventional enzyme-linked immunosorbent assay ELISA [88,89]. In the suspension assay, the capture antibodies are immobilized on colour- or size-coded microspheres that can be distinguished by their fluorescence intensity in a flow cytometer [88,89]. Each microsphere accommodates a “sandwich” consisting of the captured antibody, the target analyte and the detection antibody, which is labelled with a single chemiluminescent/fluorescent reporter. After the washing stages, lasers excite the reporters, and the emitted light is collected by a series of detectors for quantitative analysis. The use of different coloured microspheres enables the simultaneous detection of many analytes in the same sample [8].

Thousands of proteins can be quantified in parallel through the use of multiplexed immunoassays. The application of these assays has had a profound impact on a wide range of research and diagnostic areas. A number of multiplex assays have been developed for the diagnostic market. Currently, Luminex’s xMAP^®^ Technology (Luminex Corp., Austin, TX, USA) (Luminex Corp., Austin, TX, USA) is the most commonly used platform in commercial assays. It combines advanced fluidics, optics and digital signal processing with proprietary microsphere technology to provide multiplexed assay capabilities. Moreover, its open-architecture design can be configured to perform a wide variety of protein or nucleic acid assays quickly, cost-effectively and accurately [90].

Suspension microsphere-based multiplexed immunoassays have been used to analyse the expression of cytokines, chemokines and growth factors in diverse samples (serum, plasma and tissue culture) and therefore serve as a very straightforward approach for biomarker discovery [91,92,93,94,95,96]. Cytokines, chemokines and growth factors are cell signalling proteins that mediate a wide range of physiological responses, including immunity, inflammation and haematopoiesis. Changes in the levels of these biomarkers are associated with a spectrum of diseases, including autoimmune diseases such as RA [97]. Some studies have shown that the levels of signalling molecules and markers of bone metabolism circulating in the blood of RA patients could be additional markers used in assessing the degree of activity in RA [98,99]. Recently, O’Neil et al. showed that the serum proteome provides a source of proteins that serves both to identify the molecular pathways involved in the development of RA and to classify at-risk individuals [100]. The multiple biomarker approach could utilize a variety of new potential biomarkers and could be used for the early diagnosis of seronegative RA patients, the measurement of disease activity and as a prognostic tool during treatment therapy.

## 5. Conclusions

The preclinical period in RA development is characterized by immune dysregulation and inflammation without typical manifestations of the disease. During this period, classical laboratory tests (RF and ACPA) can be negative. Multiple genetic studies have revealed the genetic contribution to RA to be between 30% and 60%. The high genetic contribution indicates that genetic studies are essential to identify individuals at risk for RA and to initiate prevention measures, including pharmacological targeting and dietary and lifestyle interventions. For cases without a genetic contribution, novel and more sensitive protein biomarkers are particularly necessary.

The advancement of proteomic techniques in the last two decades has allowed the identification of a large number of candidate proteins that can act as potential biomarkers for several diseases [101,102,103,104,105,106]. This development in particular shows promise for the treatment of highly prevalent disorders, such as cardiovascular disease or cancer, in which early diagnosis is currently impeded by the absence of adequate diagnostic biomarkers. In many other diseases, such as RA, proteomic techniques represent a golden opportunity to improve the current diagnostic strategy. Multiplex immunoassays are a novel tool that allows the application of the knowledge obtained via proteomics research for the diagnosis, monitoring and treatment of diseases [84]. Classical diagnostic approaches based on single parameters have shown enormous limitations in most diseases. Only the specific characteristics of some disorders, such as diabetes, allow such approaches to remain valid. Most diseases are complex and multifactorial processes, for which a more global and ambitious approach based on the analysis of multiple protein biomarkers is required.

In contrast to the single biomarker approach, which focuses only on a particular aspect of the pathologic process, multiplexed immunoassays allow the simultaneous measurement of multiple proteins in biological samples, which has great potential for use in both basic research and clinical diagnostics. From a clinical perspective, the combination of a large number of proteins in multiplex panels would allow for the monitoring of disease states in a much more complete way, with most pathophysiological processes being evaluated at the same time. Additionally, multiplexing not only offers advantages from the point of view of the diagnosis and management of diseases but is also cost-effective. Multiplex assays could provide a more complete picture of the disease and the pathways involved, which would reduce the number of imaging tests required, prevent the repetition of unnecessary laboratory tests and reduce hospital stays, among other benefits [107,108].

Despite all these advantages, multiplex immunoassays also present distinctive challenges not encountered with single-plex assays. Companies such as Millipore, Bio-Rad, Invitrogen, Qiagen and PerkinElmer currently offer multi-protein panels for research purposes in several species, including humans. The widespread adoption of multiplexing for in vitro diagnosis will depend on the availability of robust, affordable analytical platforms and validated multiplex-optimized antibody panels as well as a high level of automation. In RA in particular, simultaneous analysis of multiple biomarkers will yield biomarker signatures of RA subtypes that will allow patients to benefit from personalized medicine.

## Figures and Tables

**Figure 1 jpm-10-00202-f001:**
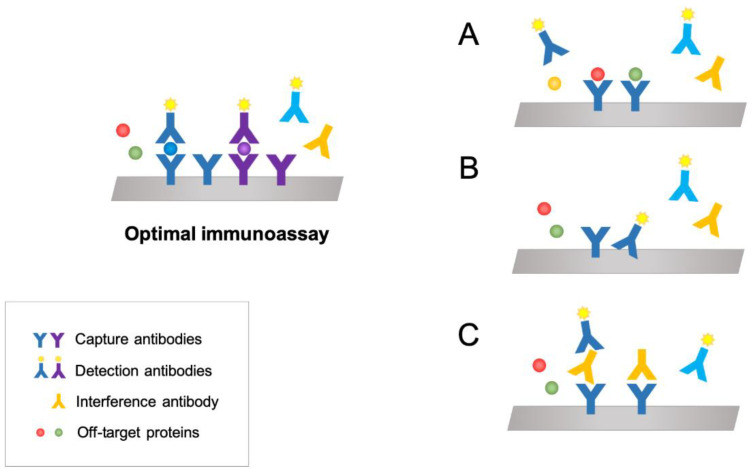
Potential sources of interference in multiplex immunoassays; (**A**) antibody cross-reactivity, (**B**) non-specific antibody binding and (**C**) antibody interference.

**Figure 2 jpm-10-00202-f002:**
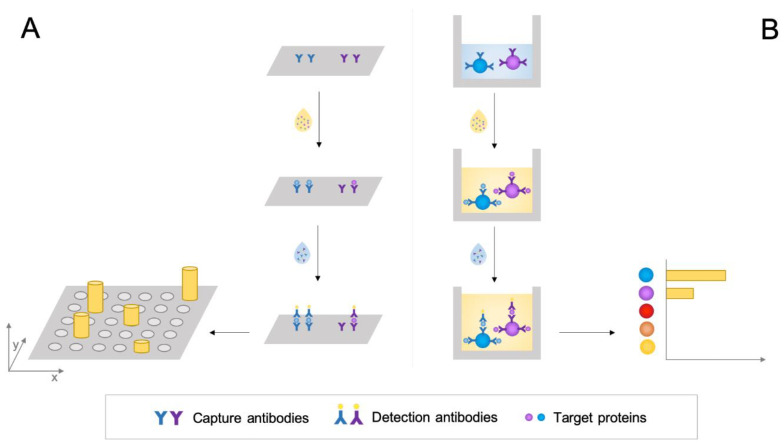
Multiplexed immunoassay systems: (**A**) planar-based assays and (**B**) microbead-based suspension assays.

**Table 1 jpm-10-00202-t001:** Environmental factors associated with the development of rheumatoid arthritis (RA).

Predisposing Factors	ACPA
Smoking	Positive
Pollution	Positive
Periodontitis	Positive
Pregnancy	Negative
Obesity	Negative
Diet	Negative
Exercise	Negative
Microbiome	Negative

ACPA: Anti-cyclic citrullinated peptide antibodies.

**Table 2 jpm-10-00202-t002:** Overview of biomarkers with potential utility in rheumatoid arthritis.

Biomarker	Comments	Reference
ACPA	Diagnostic and prognostic value. Included in the 2010 ACR RA classification criteria.	[34]
RF	Diagnostic and prognostic value. Included in the 2010 ACR RA classification criteria. Moderate specificity	[34]
Anti-CarP antibodies	Lower sensitivity than ACPAs and RF but showed promising results in combination with ACPA/RF. Associated with erosive disease.	[38]
Anti-PAD antibodies	Anti-PAD4 antibodies have been associated with radiographic progression.	[39,40]
Anti-MCV antibodies	Associated with bone erosion	[41]
14-3-3η protein	Diagnostic utility for RA. Associated with radiographic progression in early RA	[42]
Calprotectin	Associated with disease activity.	[43]

Anti-CarP: Anti-carbamylated protein; ACPA: Anti-citrullinated protein autoantibodies; ACR: American College of Rheumatology; anti-MCV: Anti-mutated citrullinated vimentin; anti-PAD: Anti-peptidylarginine deiminase; RF: Rheumatoid factor.

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
