# Peer review of "Contribution of Multiplex Immunoassays to Rheumatoid Arthritis Management: From Biomarker Discovery to Personalized Medicine"

_jpm, 2020, doi:10.3390/jpm10040202_

Round 1

Reviewer 1 Report

The article is generally well written but I would suggest that some parts are added and/or rewritten for clarity.

First comment: the article is mainly focused on use of multiplexed immunoassays as a part of the repertoire of proteomics. A short introduction and positioning of the types of proteomics would clarify things. Proteomics is nowadays largely associated with mass spectrometry based techniques, although this of course does not needs to be the case. Multiplexed immunoassays, single-molecule sequencing, gel-based techniques are all complementary and valid techniques to study the proteome, but stating which technique is the focus will make the paper more readable. This would also make the discussion on the limitations of proteomics more nuanced and avoid confusion (for example, LC-MS based DDA proteomics of plasma has well know disadvantages in clinical proteomics, but these can be partially solved by using SRM/PRM acquisition and the use of the latest generation of LC and MS systems). 

The term multiplexed proteomics in the title is confusing since the term is also used for some label based LC-MS techniques. I would use just immuno based proteomics or multiplex immunoassays.

Second comment: A quick overview of the literature concerning protein markers in RA shows that the review is focused on a few markers for the early detection or stratification of patients. The link with personalized medicine is not directly clear since the markers are (not yet?) used to classify patients or to make a choice between therapy options/outcome. For example, the anti-PAD2 marker is associated with a less severe form of inflammation but can this information be used to tailor a treatment? 

Another topic that could be described are the protein markers that are used to monitor response to therapy (like e.g. Lequerre et al., 2019: Predictors of treatment response in rheumatoid arthritis). 

These two additions would make the article more complete. 

Author Response

  1. The article is mainly focused on use of multiplexed immunoassays as a part of the repertoire of proteomics. A short introduction and positioning of the types of proteomics would clarify things. Proteomics is nowadays largely associated with mass spectrometry based techniques, although this of course does not needs to be the case. Multiplexed immunoassays, single-molecule sequencing, gel-based techniques are all complementary and valid techniques to study the proteome, but stating which technique is the focus will make the paper more readable. This would also make the discussion on the limitations of proteomics more nuanced and avoid confusion (for example, LC-MS based DDA proteomics of plasma has well know disadvantages in clinical proteomics, but these can be partially solved by using SRM/PRM acquisition and the use of the latest generation of LC and MS systems).

R: We want to express our appreciation to the reviewer for this suggestion with which we fully agree. For this reason, we have added a brief introduction to section 4 (pages 5, lines 208-215), including two new references (refs. 66 and 67).

  1. The term multiplexed proteomics in the title is confusing since the term is also used for some label based LC-MS techniques. I would use just immuno based proteomics or multiplex immunoassays.

R: Similar to the previous point, we agree with the reviewer that "multiplexed proteomics" can be confusing to the reader and for that reason we have decided to change the title to “Contribution of multiplex immunoassays to rheumatoid arthritis management: From biomarker discovery to personalized medicine”

  1. A quick overview of the literature concerning protein markers in RA shows that the review is focused on a few markers for the early detection or stratification of patients. The link with personalized medicine is not directly clear since the markers are (not yet?) used to classify patients or to make a choice between therapy options/outcome. For example, the anti-PAD2 marker is associated with a less severe form of inflammation but can this information be used to tailor a treatment?

R: Thank you for the comment. According to the Reviewer’s suggestion, we have included more information focussing on biomarkers that can stratify patient or to make a choice between therapy options/outcome. Please see revised manuscript, page 5, lines 191-206.

  1. Another topic that could be described are the protein markers that are used to monitor response to therapy (like e.g. Lequerre et al., 2019: Predictors of treatment response in rheumatoid arthritis).

R: We thank the reviewer comment. We have also included information regarding biomarkers and response to therapy (page 5, lines 191-206)

On behalf of all coauthors, many thanks for this insightful review.

Reviewer 2 Report

Overall, this is a well-written review. The identification of highly specific and sensitive bio-makers for early RA diagnosis and patient stratification treatment is a great challenge. In the context of personalized medicine, an additional paragraph focussing on biomarkers that can stratify patient in terms of clinical response (non-responders/responders to methotrexate, anti-TNF therapy, ext.) would strengthen this review.

Author Response

  1. In the context of personalized medicine, an additional paragraph focussing on biomarkers that can stratify patient in terms of clinical response (non-responders/responders to methotrexate, anti-TNF therapy, ext.) would strengthen this review.

R: We agree with the Reviewer’s suggestion. We have included more information regarding biomarkers that can stratify patient or to make a choice between therapy options/outcome. Please see revised manuscript, page 5, lines 191-206.

On behalf of all coauthors, many thanks for this insightful review.

Reviewer 3 Report

The authors Laborde et al reviews rigorously the field of the disorder Rheumatoid Arthritis focusing on available biomarkers. The authors describes the problems associated with their detectability using classical proteomics methodologies and continues describing the various antibody based assays with multiplexing capabilities.

I would like to suggest the inclusion in the publication of technologies like aptamer usage (Somascan), Proximity extension assay (Olink panels) as well as targeted proteomics (MRM/PRM) in the paper since they can be highly multiplexed and are widely used.

Author Response

  1. I would like to suggest the inclusion in the publication of technologies like aptamer usage (Somascan), Proximity extension assay (Olink panels) as well as targeted proteomics (MRM/PRM) in the paper since they can be highly multiplexed and are widely used.

R: We want to express that we fully agree with the reviewer on this suggestion. "Multiplex proteomics" can be indeed confusing to the reader as it applies to other widely used techniques, as pointed out by the reviewer. Since our goal in this review is to focus on immunoassays, we have decided to partially modify the title to “Contribution of multiplex immunoassays to rheumatoid arthritis management: From biomarker discovery to personalized medicine”. We hope this change completely eliminates any confusion for the reader.

On behalf of all coauthors, many thanks for this insightful review.
